# A Framework for Full Decentralization in Blockchain Interoperability

**DOI:** 10.3390/s24237630

**Published:** 2024-11-29

**Authors:** Seth Djanie Kotey, Eric Tutu Tchao, Andrew Selasi Agbemenu, Abdul-Rahman Ahmed, Eliel Keelson

**Affiliations:** 1Distributed IoT Platforms, Privacy and Edge-Intelligence Research Lab, Kwame Nkrumah University of Science and Technology, Private Mail Bag, University Post Office, Kumasi 999064, Ghana; asagbemenu@knust.edu.gh (A.S.A.); aarahman.soe@knust.edu.gh (A.-R.A.); ekeelson@knust.edu.gh (E.K.); 2Department of Computer Engineering, Kwame Nkrumah University of Science and Technology, Private Mail Bag, University Post Office, Kumasi 999064, Ghana; 3Department of Telecommunications Engineering, Kwame Nkrumah University of Science and Technology, Private Mail Bag, University Post Office, Kumasi 999064, Ghana

**Keywords:** blockchain gateway, interoperability, heterogenous, architecture, data integrity, peer-to-peer

## Abstract

Blockchain interoperability is one of the major challenges that has sprung up with the advancement of blockchain technology. A vast number of blockchains has been developed to aid in the continuous adoption of blockchain technology. These blockchains are developed without a standard protocol, therefore making them unable to interoperate with other blockchains directly. In this paper, we present a framework to enable blockchains to interoperate in a decentralized setup. This framework maintains the decentralized property of blockchains. Additionally, an encryption of data is employed in the transfer of data between blockchains with hash-based verification ensuring the integrity of data transferred. Light client verification, based on Simplified Payment Verification, is used as a final security measure to ensure only valid transactions go through consensus to be appended to the destination blockchain. A peer-to-peer network setup modified for use in the proposed framework is also presented. The peer-to-peer setup is tested and compared with a single client–server setup to determine the computational impact it will have when implemented.

## 1. Introduction

The development of blockchain technology has been on the rise in recent years. The rise of the cryptocurrency bitcoin [1] presented a breakthrough in data storage. Centralized servers had been the main data storage mediums in use until now. Blockchain eliminates the need to store data on centralized servers by distributing the data among peer devices while maintaining a truth state on all peer devices [2]. There have been many blockchains proposed since Bitcoin [3]; however, without a standard protocol [4] to follow in developing a blockchain, the ability for them to interoperate becomes challenging [5].

Blockchain interoperability can be viewed from two perspectives: digital asset interoperability (which deals with cryptocurrency and cryptocurrency-backed assets) and arbitrary data interoperability [6]. Digital asset interoperability has had much focus in the literature due to the numerous cryptocurrency-focused implementations of blockchain. Arbitrary data interoperability is gradually receiving attention as more data-backed use cases of blockchain are being implemented, for instance in the Internet of Things (IoT) [7], and is the major focus of this paper.

This paper presents a framework for heterogenous blockchain interoperability. The proposed framework is designed for data-based blockchain implementations and is based on improving the concept of blockchain gateways [8,9]. Decentralization is a key property of blockchains, and to maintain this decentralization, a modified peer-to-peer (P2P) setup is used for communication between blockchains. The data transferred over this communication setup is encrypted, and a hash-based verification ensures that the integrity of data transferred is maintained. As a final security measure, light client verification, based on Simplified Payment Verification (SPV), is used to ensure that only valid transactions are submitted to the receiving blockchain. A trust management service is used to monitor the behavior of participating nodes with a leader election algorithm electing a leader to perform managerial tasks on the P2P network. The main contributions of this paper therefore include the following:A framework for decentralized blockchain interoperability is presented.A modified peer-to-peer (P2P) setup is presented as a communication link between two blockchains.The proposed P2P setup is tested to determine its computational impact.

The rest of this paper is organized as follows: Section 2 presents the literature review, and Section 3 presents our proposed architecture. Section 4 discusses the security considerations and theoretical validations of the architecture, Section 5 presents the results of testing the P2P setup, with Section 6 concluding this paper and presenting future works.

## 2. Literature Review

The problem of blockchain interoperability has caught the attention of researchers. Several authors have proposed solutions to address this problem. The three major classes of blockchain interoperability solutions are notary schemes, hashed time lock contracts (HTLCs) and relays. Of the three, relays are best suited for more complex use cases, including cross-chain data movement operations [10]. Relays have seen a significant amount of research activity relative to notary schemes and HTLCs, and they are seen as the future for enabling fully decentralized interoperation between blockchains [11]. We discuss proposed works in the literature in this section.

### 2.1. Notary Schemes

Notary schemes use the concept of notarization to enable interoperability between blockchains: trusted nodes ensure execution of the interoperation action [10]. Scheid et al. [12] proposed an API implementation called Bifröst, which works on an abstracted layer above the interoperating blockchains. Bifröst is modular and uses interfaces and pluggable adapters that connect to the different blockchains to enable data storage and retrieval of data. Gao et al. [13] proposed a cross-chain data migration scheme that employs an oracle responsible for managing data transfer requests between blockchains. Their proposed solution takes advantage of smart contracts to trigger the data transfer process. Punathumkandi et al. [14] proposed the Fabreum infrastructure, which uses a notary to verify the occurrence of an event on a blockchain. A decentralized application establishes connection between communicating blockchains by storing blockchain nodes as interaction endpoints. The Cross-Blockchain Communication Protocol (CBCP) and Cross-Chain Communication Protocol (CCCP) establish the communication rules for interoperation.

### 2.2. HTLCs

HTLCs enable interoperability by providing a method of locking assets on a source blockchain and releasing the asset after the destination blockchain indicates its readiness to receive the asset [15,16]. Zamyatin et al. [17] proposed XClaim as a generic framework to enable trustless cross-chain exchanges. A one-to-one mapping is conducted between blockchain-based assets and cryptocurrencies, which can then be swapped between blockchains atomically. Wanchain [18] was proposed as a platform to enable cryptocurrency exchange between blockchains. Validator nodes validate cross-chain transactions, which are then forwarded to the main Wanchain platform. Multi-party computing and threshold secret-sharing technologies are used to manage accounts without a third party. Multi-party computing and threshold secret-sharing technologies are used to manage accounts without a third party. Barbàra and Schifanella [19] proposed MP-HTLC, a multiparty implementation of hash time-lock contracts. They made use of multiparty computation to create a secret for multiple participants instead of having a single participant from either blockchain. Puneeth and Parthasarathy [20] proposed a cross-chain interoperability method using an intermediate smart contract network. A user on a destination chain requesting data from a source chain submits the request, and a hash lock is created for that user. The hash lock involves switching to the intermediate contract network, retrieving the network address of the source blockchain and switching to that network, retrieving the contents, reverting to the original destination contract and returning the data.

### 2.3. Relays and Relay Chains

Relays provide the most direct method of interoperability between blockchains. This class on interoperability is currently the most actively researched method of interoperability due to the fact that all operations are managed and executed by the participating blockchains only. Blockchains provide some information about their ledgers to other blockchains, and this information makes it possible to verify a cross-chain transaction request originating from the former blockchain [10,21]. Relay chains use an additional blockchain (master blockchain) to enable this operation. A BTC relay [22] was implemented as a set of smart contracts to verify transactions on Bitcoin blockchain for cross-chain transactions with Ethereum blockchain. A light version of Bitcoin blockchain is maintained on Ethereum, which is used to verify transactions originating from Bitcoin. Polkadot [23] was proposed to enable interoperability using the concept of blockchain bridges. A bridge relay performs consensus before a cross-chain transaction is executed. A bank, which is a group of actors, provides a way for sending or receiving cross-chain assets.

Overledger [24] is a proposed blockchain operating system that allows applications to be executed cross-chain. Overledger abstracts the underlying blockchains to provide a layer of abstraction for interoperation. A messaging layer is responsible for retrieving and storing information received from the various underlying blockchains, and a filtering and ordering layer relays the messages received from each blockchain. HyperService [25] provides an abstraction layer for interoperating decentralized applications. Blockchain drivers, known as verifiable execution systems, translate programs into blockchain transactions executable by the blockchain. A universal inter-blockchain protocol (UIP) manages complex cross-chain transactions securely and atomically. Testimonium [26] is proposed as a validation-on-demand relay scheme. An incentive scheme is used to regulate the behavior of participating validators. A section of validators is also responsible for identifying and disputing invalid block headers of the source blockchain. Tesseract [27] was developed as a real-time cryptocurrency exchange service. Trusted hardware is used to set up the relay, and operations are executed in a trusted execution environment. Jin et al. [28] presented a multi-blockchain interoperability architecture. The architecture operates in either one of two modes: an active mode where a blockchain waits for a response after submitting a request to a second blockchain, and a passive mode where a blockchain monitors a second blockchain for events and transactions.

Fynn et al. [29] proposed the Move protocol, which enables the movement of objects and smart contracts between blockchains. The state of a smart contract is locked on a source blockchain and reconstructed on the destination blockchain after proof of the state is determined with a Merkle proof. Hermes was proposed by Belchior et al. [30] as a middleware to enable blockchain interoperability. Hermes was developed to enable a gateway connecting two blockchains to transfer assets across using an extension of the Open Digital Asset Protocol. Gateways are nodes residing on each respective blockchain and are aware of the other gateways either directly or indirectly through a gateway register. Darshan et al. [31] proposed MainChain, a concept that connects blockchains together via a central chain. Data can be sent to a second blockchain via mixed nodes. A transaction that was added to the first blockchain is forwarded by the mixed nodes to the central chain, which then forwards it to the second blockchain. Karumba et al. [32] proposed BAILIF, a blockchain agnostic interoperability framework. The proposed framework makes use of a bridge, which consists of trusted nodes in the participating blockchains to form a notary service. A notary service node receives a cross-chain transaction request from the application layer and a request is made for the transaction to be endorsed on the source blockchain. The source notary nodes then send the transaction together with proof of its existence to the destination notary nodes. These nodes then submit the transaction to their respective blockchains.

The development of notary schemes has seen a drop mainly due to the trust requirement of the notary. HTLCs, on the other hand, are designed mainly for cryptocurrency exchange and are not feasible for data-based use cases. Relay schemes are used for both cryptocurrency and data-based use cases, and a significant amount of development regarding blockchain interoperability is taking place in this area. The relay solutions proposed in [28,30] did not implement or test their proposed solutions. Polkadot Bridges [23] does not require any modifications to existing blockchain frameworks; however, it is not fully decentralized and is designed to work with specific blockchains. BTC relay [22] is fully decentralized; however, it only works with cryptocurrency exchange between Bitcoin and Ethereum blockchains. The only other fully decentralized solution proposed was the Move protocol [29]. It is also designed to work with cryptocurrency-based blockchains and requires some existing blockchains to be modified to accommodate the design. Testimonium [26] and Tesseract [27] require no modification to existing blockchains; however, they are not fully decentralized, and they are not designed for data-based blockchains. Overledger [24], Hyperservice [25], MainChain [31] and BAILIF [32] target data-based blockchain interoperability; however, not all of them are fully decentralized. Additionally, Overledger is not blockchain agnostic. We tackle these shortcomings by proposing a fully decentralized solution that is blockchain agnostic.

## 3. Proposed Framework

This section presents a decentralized blockchain interoperability framework for data migration. The proposed framework enables data to be moved between heterogenous blockchains across a P2P network. The major requirements considered in developing the system were the following:Trustless: the proposed system should not rely on trusting any single entity to enable the success of interoperability. Blockchains work without trusting any single entity in the network. Eliminating the need for trust ensures that the data transferred between blockchains can be verified and validated by multiple nodes, and the destination blockchain does not depend on the trustworthiness of a single node.Decentralized: the proposed system should be built on a decentralized platform to ensure no single entity can dictate how interoperation is achieved. Furthermore, a decentralized system ensures multiple nodes participate in interoperation, making it easier to detect malicious nodes.Blockchain Agnostic: the proposed system should work between heterogenous blockchains without relying on their underlying frameworks. This would make it cross-blockchain compatible.Accountability: any action taken in the system should be traceable to the acting node. If a node acts maliciously, the actions of that node should be attributed to it, and a penalty given to that node. This requirement will help control the behavior of nodes in the network.Fault Tolerant: the proposed system should be able to continue functioning after the failure of some nodes. Multiple nodes should participate in the data transfer process to ensure that if some nodes fail, there will be other nodes to complete the process.

Drawing from the requirements stated above, we base our architecture on the concept of blockchain gateways presented by Hardjono et al. [8,9,33]. Each blockchain is seen as an autonomous system with a gateway node. Gateways act as dual-facing nodes in the blockchain network, capable of interacting with nodes inside the network and externally with gateway nodes on other blockchains. The concept of gateways has the potential of creating an internet of blockchains [34]. The proposed concept in its current state introduces centralization and the need for trust, with trusted nodes acting as gateways for each blockchain, and a failure of such a node cuts off the connection to the blockchain it represents. We seek to provide an improvement on this design.

The proposed architecture is shown in Figure 1. We assume that Blockchain A has its own ledger and a blockchain network that is different from Blockchain B’s ledger and network. We also assume that at least one of the blockchains is a permissioned blockchain.

The architecture has BC ‘A’ as a permissioned blockchain and BC ‘B’ as a public blockchain. BC ‘A’ has an enrollment service that enrolls participants on the blockchain and assigns roles to the participating nodes. The roles determine what data each node can access and what actions they can perform. The enrollment service is also used to create Light Clients on BC ‘B’. Light Clients on BC ‘B’ maintain the block headers of BC ‘A’ to verify transactions on BC ‘B’ originating from BC ‘A’. The encryption algorithm encrypts the data that will be sent over the peer-to-peer (P2P) channel to secure the data in transit. The encrypted data are then hashed, which will be used to validate the data received by BC ‘B’ nodes. The encrypted data together with the hash are stored in the Shared Space, a persistent data storage space accessible by both blockchain and P2P applications. The Meta-info Generator creates a meta-file for distribution by the tracker in the P2P communication. Uploaders (UPs) are BC ‘A’ nodes, which maintain the full blockchain ledger. These nodes share the data securely over the P2P network. The Tracker (TR) is also a participant on BC ‘A’ and is elected by a Leader Election Algorithm. It distributes the meta-file in the P2P network. The Downloaders (DLs) are full participants of BC ‘B’ and act as light clients on BC ‘A’. These nodes enable relay interoperability. The data received over the P2P network are stored in the Shared Space. The Downloaders verify the data with the BC ‘A’ block headers by performing a Merkle proof on the transactions. The Merkle proofs are then validated by BC ‘B’ nodes, and the data are decrypted and submitted to BC ‘B’ for consensus, validation and appending to the ledger. The Trust Management Service provides trust values, which are used to monitor the behavior of DLs and by the Leader Election Algorithm to elect a tracker.

The system works by transferring insensitive data, i.e., data that can be made available to participants in a second blockchain, from the source blockchain to the destination blockchain. Transactional data not labeled as sensitive data are retrieved and encrypted independently by each UP node. The encrypted data are hashed, and the hash value together with the encrypted data are stored on a section of the UP node’s storage, called the BC-P2P Shared Space for onward distribution over the P2P network. The BC-P2P Shared Space is a reserved location on local storage accessible to both Blockchain and peer-to-peer applications. This storage location ensures the blockchain storage location with its data is kept secure and directly inaccessible to other applications. The TR node generates a meta-info file from the data retrieved from the blockchain. This meta-info file contains information that includes the initial set of UP nodes available for DL nodes from which to receive data. The TR sends the generated meta-info file to DL nodes. The DL nodes, on receiving the meta-info file, connect to available UP nodes over the P2P network. The UP nodes transfer the encrypted data and the hash value to DL nodes. The DL nodes generate a hash of the received data and compare them with the received hash value. A match in hash values is an indication that the data were not corrupted or modified in transit. The received data are then decrypted and stored in the BC-P2P Shared Space. Transactions are then verified via light client verification. This verification determines if the transactions are valid transactions that indeed exist on the source blockchain. The BC ‘A’ headers maintained by DL nodes are used, together with a Merkle path for each transaction, to perform a Merkle proof. Successfully verified transactions are forwarded for consensus and appending to the BC ‘B’ ledger. The trust value is calculated after each data migration session based on the behavior of each node; however, each node is given a full trust value before the first session. The TR node (leader) is elected before the first session, and a new leader is elected when the trust value of the current leader reduces or when the leader goes offline for an extended period.

A layered view of the architecture is presented in Figure 2. There are three levels in a node over which the architecture spans. For each node, there is the blockchain level, local device environment level and the network level. The blockchain level shows participation of the node in a blockchain. The tracker, which is a participant on the source blockchain, generates a meta-info file from the retrieved transactions on the ledger. Only transactions that have been validated and appended to the blockchain ‘A’ ledger are transferred to blockchain ‘B’. Uploaders are participants of blockchain ‘A’ and maintain a full copy of the ledger. Downloaders are participants of blockchain ‘B’, maintain a full ledger and are also participants of blockchain ‘A’ but maintain just the block headers of the blockchain ‘A’ ledger. The local device environment is localized to each node. Hash values are calculated and an encryption or decryption of the data is performed within each node. The Shared Space is a location where the data are handed off between the blockchain application and the P2P application. This hand-off keeps blockchain data secure from the P2P application, making only relevant data available to the P2P application. The network level is where the P2P application works. A P2P network is created to enable communication between nodes for data transfer. External services are the Leader Election Service and the Trust Management Service. The Leader Election Service elects a node residing on blockchain ‘A’ to act as the leader (tracker). The leader is elected based on information submitted by the Trust Management Service. The Trust Management Service monitors the nodes in the network and assigns a trust value to each node based on their actions in the network.

Figure 3 and Figure 4 present the interaction diagram of the system. The interaction diagram describes communication between a single tracker, uploader and downloader. It describes in detail how communication is achieved sequentially between nodes. In Figure 3, Downloader nodes are enrolled onto blockchain ‘A’ and given the privilege of storing the block headers. Before a data migration session begins, the data are encrypted, the meta-info file is distributed and connections are established between uploaders and downloaders. Downloaders can connect to multiple uploaders to receive data, and uploaders can also connect to multiple downloaders and serve them with data. Nodes submit an updated list of connected peers to the tracker node. This list is used to keep track of which nodes are communicating. Downloader nodes also submit a list of received portions of data to the tracker node. This list is used to trigger the Rarest Piece First Algorithm (RPFA) [35,36]. The RPFA keeps a balanced distribution of data and is executed when portions of data become scarce in the network. The End Game Algorithm [35,36] is executed when a Downloader node has received most of the data. Figure 4 shows the interaction between the actors from the execution of the End Game Algorithm onwards. The algorithm speeds up the receiving of the data by sending requests for the remaining portions of the data to multiple nodes simultaneously. This action ensures that a portion of the data is received as early as possible from the first available node. After the data are received by the downloader node, the data are decrypted, and the node performs the light client verification then submits the data to blockchain ‘B’.

We achieve the requirements considered for designing the architecture below:Trustless: the use of a decentralized interface between the communicating blockchains ensures that no single node needs to be trusted. Validating by hash value and verifying data via light client verification by multiple nodes means that the operation of the system and the data being transferred do not depend on the trustworthiness of a single node.Decentralized: a P2P network provides a decentralized interface of communication between blockchains. The decentralized interface eliminates the need for having a centralized entity managing the interoperation process. This decentralized interface also makes the system more robust to attacks that target centralized systems.Blockchain Agnostic: the concept of blockchain gateways enables nodes to communicate without depending on the underlying blockchain frameworks. Communication between blockchains occurs at a level above the blockchain, removing dependency from the blockchain itself. This way, the data are retrieved and sent from the source blockchain, verified by the destination blockchain nodes and submitted to the destination in a way that is similar to how regular transactions are submitted.Accountability: a trust management service ensures that the actions of participating nodes are monitored and every action can be attributed to a particular node. Interactions between nodes are reported and offending nodes are given a penalty in the form of a reduction in trust score.Fault Tolerant: the participation of multiple nodes in the P2P network ensures that there is no single point of failure in the system. The failure of some nodes will not cause the system to fail. Multiple nodes in the network means that there are nodes always available to transfer and other nodes to receive data.

### 3.1. Peer-to-Peer Communication

The concept of blockchain gateways proposed by Hardjono et al. [9] presented blockchains as autonomous systems with a gateway connecting them. Blockchains are decentralized by nature, and having a centralized link between them introduces a single central point of failure. Blockchains were also introduced to enable collaboration without the need for trust; however, a single link between blockchains requires the need to trust the gateway nodes. We therefore take the concept of gateways further and propose having a decentralized interface between them. Here, we discuss using a peer-to-peer (P2P) interface.

P2P communication for data sharing has been in existence for years and is seen as the most practical method of creating a decentralized platform to enable data sharing [37]. P2P has faced a lot of challenges in the past mainly due to internet ‘piracy’ [38]; however, the concept has also been applied successfully in other applications [39]. In our proposed solution, we implement a P2P communication channel to enable blockchain nodes on different blockchains to communicate with each other. The P2P network has three actors: Uploaders, Downloaders and Tracker(s). Uploaders are nodes in the source blockchain that have the data, i.e., they are full nodes on the source blockchain. They are responsible for sharing the data with nodes on the destination blockchain. Downloaders are full nodes on the destination blockchain and light nodes on the source blockchain. They connect to Uploaders to receive data from the source blockchain and verify the data via Merkle proofs (more on this in Section 3.6). The Tracker(s) is responsible for performing managerial tasks in the P2P network. Depending on the size of the network, more than one tracker may be required in the network to ensure performance of the network is at an optimal level. It creates the meta-info file to be distributed, maintains an active list of participants in the network and also maintains a list of the portions of data that the downloaders have received. This latter role is what enables the network to respond when some portions of data have been received by very few nodes and there is a possibility of incomplete receipts of data due to the scarcity of those portions of data in the network. Two algorithms, Rarest Piece First Algorithm and End Game Algorithm [35,36], optimize the delivery of data during communication.

A partially managed P2P network [38,40] is used in the proposed framework, and this network requires selecting a leader to act as the tracker. The selection of this leader is discussed in subsection C. Partially managed networks provide a decentralized method of communicating while also providing a client-server architecture. The client-server architecture ensures that nodes easily find other participants in the network to connect to without costing much overhead in the network searching for other nodes. This architecture enables nodes to also know which other nodes are available to connect to. Additionally, the location of resources can be obtained from the leader of the network. Nodes do not receive data from the leader but from other nodes in the network. This requirement decentralizes the distribution of data in the network.

### 3.2. Trust Management Service

In most online communication, trust is required to ensure better communication and a reduction in malicious activity [41]. Blockchains generally are trustless systems and do not depend on the trustworthiness of individual nodes. Rather, consensus replaces the need for trust. When two autonomous blockchain systems communicate, some nodes are given the task of enabling this operation. In some cases, a node (or set of nodes) is assigned the full responsibility. If the node or nodes turn malicious, the system is compromised. Another method is to use a dedicated blockchain to ensure that the data moving between the interoperating blockchains are validated by members in the dedicated blockchain. This system also requires some level of trust placed on the nodes submitting data to both the destination blockchain and the dedicated blockchain. Our architecture eliminates the need to trust a single node by using a P2P mode of communication. This mode of communication provides multiple data paths with multiple source and destination nodes and eliminates the need to trust a single node. The role of the TMS system in our architecture is to assist with selecting a leader (or set of leaders in a large network) to ensure that the network management tasks are performed efficiently. Additionally, it ensures that the nodes participating in the interoperation task act truthfully. Due to the decentralized nature of communication, any attempt to alter the data being transmitted will be discovered by the destination nodes as they compare the data received. This discovery will translate into a penalty applied to the offending node in the form of reducing its trust value. If the trust value falls below a threshold, the node is taken off the network. A reputation-based system will be used for the TMS. Reputation-based systems build a reputation for each member in the network based on their past activities and interactions with other members on the network.

### 3.3. Leader Election Algorithm

A leader election algorithm in a distributed system enables the nodes in the system to collectively agree on selecting a node to act as a leader [42]. As part of the proposed framework, there exists a tracker, which acts as a leader for communication in the P2P network. This tracker performs organizational tasks to ensure downloader nodes can easily locate uploader nodes. There exist several possible candidates that can act as a leader in the network, so there needs to be an election to elect one of these nodes. The TMS provides values of nodes, from which candidate nodes are placed in a pool for election. There are several algorithms in the literature for leading an election, including the Bully Algorithm, Paxos and Raft. We consider using a Raft Algorithm [43] in this architecture.

The Raft Algorithm is an alternative to the Paxos Algorithm and is used in some blockchains for consensus. It provides the use of randomized timers for leader elections, which helps resolve election conflicts quickly. It separates the main elements used in consensus and reduces the number of states that are considered in the election process. More importantly, it has several open-source implementations.

### 3.4. Encryption/Decryption Algorithm

Data encryption provides security for data in transit over a communication channel. In encryption, the data are translated into a different form by adding a secret key and passing it through an encryption algorithm. To view the original data, a decryption key together with the encrypted data are passed through the algorithm. This ensures that only the parties with the respective keys can view the data, minimizing the risk of data attacks performed on data while in transit over the communication channel.

The data transmitted over the P2P network is encrypted to prevent data attacks. Elliptic Curve Cryptography (ECC) was considered for use in our architecture. Compared to Rivest–Shamir–Adleman (RSA) and Digital Signature Algorithm (DSA), a smaller number of parameters are used, while providing a similar level of security [44]. This approach makes ECC computationally faster, power efficient and low on storage consumption [45]. For our architecture, this method will reduce the overhead used to secure data before transmission.

### 3.5. Hashing Algorithm

Data hashing is a technique that can be used to quickly verify the integrity of data. This quick verification is made possible by transforming the data into a fixed-length string of values, known as a hash, which is a representation of the data [46]. The transformation occurs when the data are passed through a hashing formula. The formula is written such that the output is the same length, regardless of the input data length. Additionally, running the same data through the formula will always result in the exact same hash, and changing a single character in the input data returns an entirely different hash output. This property of hashing formulas enables us to verify the integrity of data. A hash of the data is generated from the source before transmitting the data. The receiving entity generates a hash of the received data and compares this hash with the hash from the source. If the hash values match, then the data have not been altered.

There are several hashing algorithms that have been proposed and adopted over the years. The most popular and widely used are Message Digest 5 (MD5) and Secure Hash Algorithms 1, 2 and 3 (SHA-1, SHA-2, SHA-3). MD5 and SHA-1 are both known to be vulnerable to attacks, but they are used in some applications due to their efficiency [47]. SHA-2 is more secure than MD5 and SHA-1, even though it shares a similar internal structure as the two. It is, however, not as efficient as the MD5 and SHA-1. SHA-3 was proposed in 2012 as an alternative to SHA-2 using a different internal structure known as the sponge construction. In hardware implementations, SHA-3 is known to perform faster than MD5, SHA-1 and SHA-2.

We considered using KangarooTwelve [48], an extendable output function of SHA-3. KangarooTwelve incorporates a built-in parallel mode by using a tree hash mode to exploit multi-core architectures. It also reduces the computational effort in half by tuning the permutations required.

### 3.6. Light Client Verification

The bitcoin whitepaper introduced the concept of Simplified Payment Verification (SPV) [1]. SPV enables a transaction on the blockchain to be verified by a node with the block headers of the blockchain. A light client maintaining just the block headers of the blockchain can verify the existence of transactions on the blockchain via a Merkle proof.

The concept of SPV [49] is possible because of the use of Merkle trees in blockchain. A Merkle proof is used to verify the existence of a transaction in a Merkle tree. To perform a Merkle proof, a Merkle path of the transaction is required, together with the root of the Merkle tree. The Merkle root is obtained by recursively hashing and concatenating transactions in the tree until one final hash is found, and this root is stored in the block header. The Merkle path for a transaction is the minimum set of hashes in the Merkle tree required to reconstruct the Merkle root by recursively hashing and concatenating, and it is also known as the authentication path. Compared to maintaining the entire blockchain for transaction verification, the number of hashes necessary for computing the Merkle proof scales logarithmically. Given a number of transactions N in the tree, the number of hashes required for a Merkle proof is log2N [50].

## 4. Security Considerations and Theoretical Validation

This section discusses the security considerations taken into account when designing the architecture. Additionally, the theories applied in designing the architecture are described and justified.

### 4.1. Security Considerations

In this section, we discuss the security attacks we considered most likely to target blockchain interoperability and how our architecture manages these attacks. The security attacks listed here are attacks that, if successful, will impact the interoperation process. We exclude attacks that are not relevant to our system, for example, a double-spend attack, which is an attack that targets cryptocurrency-based systems.

DoS/DDoS Attacks: DoS/DDoS attacks attempt to disrupt the normal operation of a system by causing some nodes to go offline by flooding them with requests they cannot handle, causing them to crash. Flooding the node responsible for enabling interoperability will cause no actions to be taken as the node will be offline, preventing the transmission of data across blockchains. The P2P method of communication ensures that the flooding of some nodes will not affect the entire data transfer process, making the system resilient to these attacks.Sybil Attacks: a Sybil attack is an attack where an attacker creates multiple identities on the network to subvert the reputation system provided by the network. This attack is usually a pre-cursor to another attack. By managing multiple identities, the attacker can gain a larger influence on what occurs in the network. The use of light clients ensures that the participating nodes on the public blockchain are enrolled, thereby assigning a unique identity to all participants on the P2P network. Additionally, the TMS used in the architecture monitors the behavior of the nodes in the network and assigns them trust values. Nodes that exhibit malicious behavior have their trust value reduced.Single Point of Failure: A single point of failure in a system is a component of the system that prevents the entire system from functioning when that particular component stops functioning. The concept of blockchain gateways uses a node in a blockchain to connect to gateway nodes in other blockchains. A failure of this node means no communication between that node’s blockchain and other blockchains. We use multiple node gateways for communication to eliminate this single point of failure. The nodes in our architecture act independently, and a failure of one node does not affect the operation of the system. The use of a P2P network leader can also be a point of failure; however, a new leader is elected each time the leader fails, ensuring there is always a leader to perform the network managerial tasks.Man-in-the-Middle (MITM) attack: An MITM attack is where an attacker is positioned between two communicating parties and either listens in on their communication or actively attempts to subvert their communication. In this communication, an MITM may attempt to modify the data being transferred or inject malicious data as part of the data being transferred. The introduction of data encryption ensures that the data will be secured from modification in transit. Additionally, data verification by comparing hash values ensures that malicious data are not introduced in transit. As a final check, light client verification verifies the existence of all transactions received by the destination blockchain in the source blockchain. Any transaction that cannot be verified as existing on the source blockchain is discarded. This security feature ensures that only valid transactions are appended to the destination blockchain.

### 4.2. Theoretical Validation

This section describes the theories applied in designing the proposed architecture and why they were used. Blockchains have certain properties that need to be maintained when developing extensions to the technology. Immutability, decentralization and consensus are three main building blocks of blockchain that are key to ensuring the security of blockchain. These core properties are maintained in each blockchain as the data received by the destination blockchain are agreed on through consensus before being appended to the ledger. Additionally, confidentiality, integrity and availability are three major principles required for the transmission of data over a communication medium to ensure security. Blockchain interoperability involves moving data between two autonomous blockchains; therefore, the principles of securely transmitting data over a communication channel must be considered in addition to the core blockchain building blocks. We discuss the theories applied to ensure that we maintain the fundamentals of interoperating blockchains. Additionally, we prove the scalability of the system.

Confidentiality: data confidentiality protects the data from unintentional or unlawful access. The data being transferred over a communication channel needs to be secured to ensure unauthorized access. Any data that are not secured could lead to third parties performing other attacks like message modification or data injection attacks. To secure the data in transit, encryption is used. This security feature protects the data from being modified or false data being injected in transit. A lightweight encryption algorithm ensures that the data are secured before transmission whilst keeping the overhead in the system low.Integrity: data integrity maintains the accuracy and consistency of data over their entire usage. This security feature ensures that the data are not corrupted intentionally or unintentionally whilst in transit. A hashing algorithm creates a hash of the data before transmission. A new hash of the data is then generated and compared with the original hash, and if it matches, the data are valid. Hashing is used in our architecture to ensure that the data remain consistent as they are transferred from the source blockchain to the destination blockchain.Availability: data availability ensures that the data are readily available to users and applications when they are required. Having many sources of data distributed over the network ensure that the data are always available. A P2P network ensures that there are multiple nodes with data and that the failure of one node does not affect availability of data over the network.Decentralization: decentralization ensures that no single entity has sole authority in a system. Blockchains are inherently decentralized, and this trait ensures that no single node can dictate what occurs in the network. We maintain that property by using a partially managed P2P network as the interface between the two blockchains. This network ensures that no single entity can take over and dictate how the data move through the network. The network leader that we use in the partially managed network is only responsible for network management tasks to make communication more efficient and therefore cannot dictate how the network operates. This feature keeps the network decentralized whilst enabling an efficient communication process.

## 5. Peer-to-Peer Setup Evaluation

The P2P setup was implemented and tested with eight identical workstations consisting of an Intel^®^ Core™ I5-10500T CPU @ 2.3 GHz and 8 GB of RAM running Linux Ubuntu 22.04 sourced from Kumasi, Ghana. The workstations were all connected over a local area network (LAN). Five test scenarios were created for the P2P evaluation test: single-channel direct, multi-channel direct, single-channel encrypted, multi-channel encrypted and multi-channel filesplit. All single-channel scenarios involved just two devices, one client and one server, whilst the multi-channel scenarios involved all eight devices, four of which had data to serve and the other four receiving the data. Each of the devices also had multiple communication channels ensuring that they could connect to and either receive from all serving devices or serve all receiving devices. The encrypted scenario involved an encryption of data over the communication channel with hash-based verification. The filesplit scenario involved the files greater than 100 MB being split into smaller chunks before transferring. All test scenarios were written in Python. In total, 22 files were transferred in each scenario. Different file types were used in the tests including .JPG, .PDF and .Bin files. The smallest file size was 108 KB, and the largest file was 1.9 GB. The measurements for the graphs were split in two, one for small-sized files and a second for large-sized files. The total transfer times for each file and average transfer times for each scenario were measured. The aim here was to determine how much time is traded in favor of decentralization and added security. Most of the discussion here therefore looks at the multi-channel scenarios.

The total transfer time was measured as the average of the total time it took for each receiving device to receive the file. Figure 5 shows the time it took to transfer each small file for each given scenario. The single-channel direct scenario predictably took the least time because there were only two devices using up the entire communication bandwidth. Both single-channel scenarios were also nearly steady with increasing file size. It is important to note that the single-channel communication introduces centralization and its associated drawbacks. Looking at the multi-channel scenarios, there was an inconsistent pattern in transfer times. This can be attributed to how file requests are randomized and served. Looking at the encrypted scenarios, there is a significant increase in time in milliseconds spent compared to non-encrypted scenarios in most cases. This trend is attributed to the overhead cost of encryption.

Figure 6 shows the transfer times for large files. Here, the overhead cost of encryption reduced drastically on the time spent. This reduction can be seen when looking at the single-channel scenarios, where the time spent in all situations is near identical. The multi-channel scenarios exhibit the same inconsistent times; however, the filesplit scenario tends to take up the most time here.

Figure 7a averages the time taken to transfer a small file. The difference in time between the encrypted and non-encrypted scenarios is about 38 ms on average, which, for a non-real-time application with added security, is acceptable.

Figure 7b shows the average time taken to transfer a large file. Here, we clearly see the near identical time it takes for the single-channel scenarios, making a strong case for the use of encryption even in interoperability solutions using single gateways for data-intensive applications. The difference in time between the multi-channel direct and multi-channel encrypted is approximately 4 s, which is, for the most part, acceptable in non-time critical situations. The filesplit scenario took the most time, about 10 s more than the encrypted scenario.

Figure 7c presents the total time it took to transfer all the files. For single-channel scenarios, it is the time taken to receive all the files by a single device. For multi-channel scenarios, it is the average of the total times for all four devices to receive the files. Here too, the differences between the encrypted data transfer and non-encrypted is very small in the single-channel scenarios, because the multi-channel scenario adds on less than a minute for all 22 files sent.

Blockchains generally are not implemented as part of systems where response time is of critical importance. Arriving at consensus and creating a block to distribute to other peers on the blockchain and update the ledger are not ideal for time-critical applications. Such applications use a database for time-critical operations and the blockchain for long term storage. Ensuring security on the blockchain is of higher importance than the time it takes. However, to eliminate the overhead cost of encryption, data can be stored in encrypted form at the source, which would further reduce the time it takes to transfer data. These tests have further proven the feasibility of having a P2P communication medium with added security in developing a blockchain interoperability solution. The added security and, importantly, the decentralization it provides, make up for the time spent.

The case for file splitting could be made for situations where bandwidth is limited in data intensive applications to reduce the time it may take to recover from failures associated with low bandwidth data transfers. In this paper however, this scenario is not deeply investigated and could be an area of focus for applications being used in very low bandwidth situations.

## 6. Conclusions and Future Work

This paper has presented a heterogenous blockchain interoperability solution for data migration. The concept of blockchain gateways was enhanced to propose a decentralized interface to interoperate heterogenous blockchains. In the proposed framework, a P2P setup is used for communication between blockchains and light client verification to verify transactions received from the source blockchain at the destination blockchain. The data are encrypted before transmission over the P2P network, and the integrity of the data are maintained via hash-based verification. A trust management service is used to monitor the behavior of participating nodes with a leader election algorithm electing a leader to perform managerial tasks on the P2P network. The proposed solution was theoretically validated and the P2P setup was tested and evaluated to determine its impact when implemented.

The proposed framework will be implemented and tested as part of future works. The performance of the interoperation process will be measured and compared to previous proposed solutions. The current state of the proposed framework relies on a Merkle proof to verify the existence of data on the source blockchain. For blockchains that do not use a Merkle tree structure, there is no trivial method to verify the existence transactions originating from them. A future direction could be designing a transaction verification method for such blockchains.

## Figures and Tables

**Figure 1 sensors-24-07630-f001:**
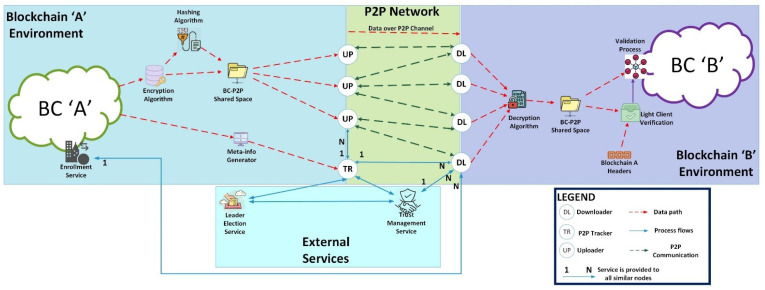
Proposed architecture.

**Figure 2 sensors-24-07630-f002:**
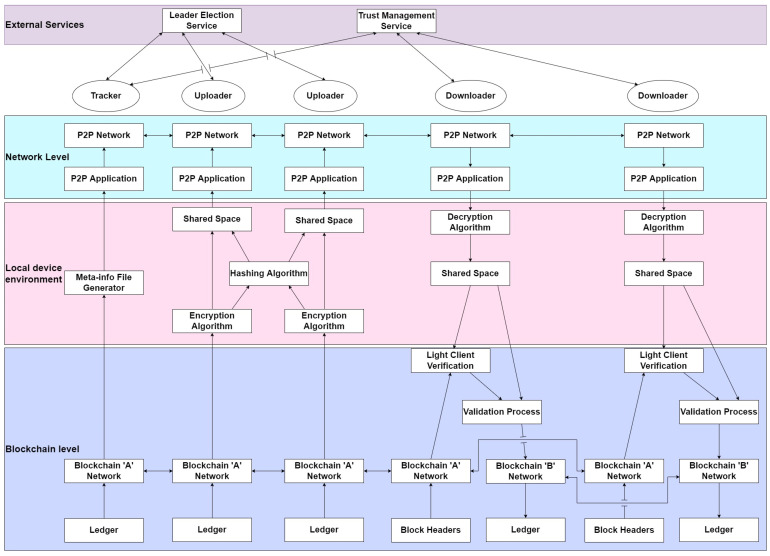
Layered view of the architecture.

**Figure 3 sensors-24-07630-f003:**
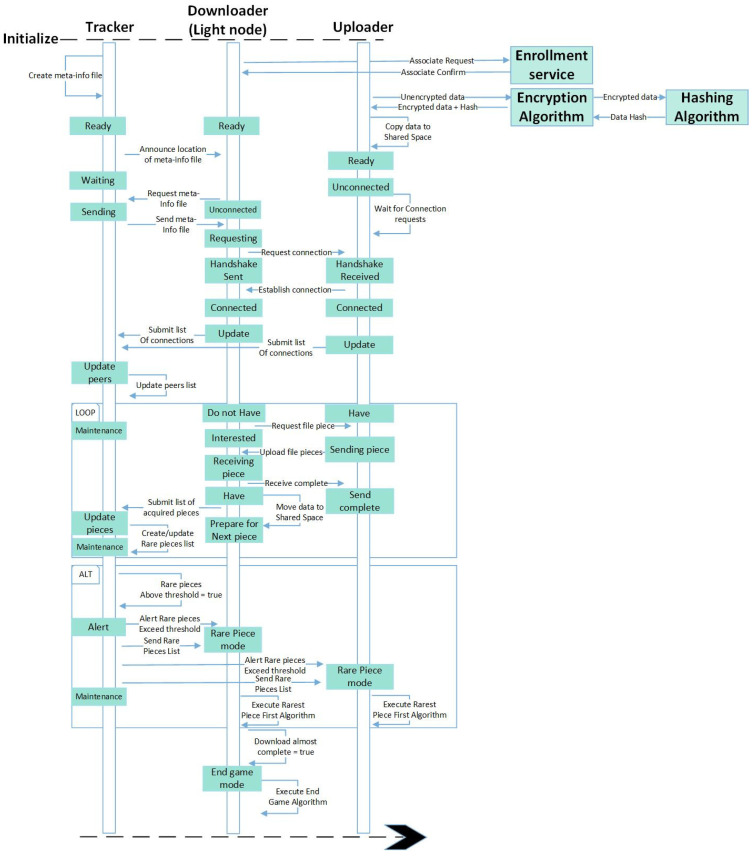
Interaction diagram part 1.

**Figure 4 sensors-24-07630-f004:**
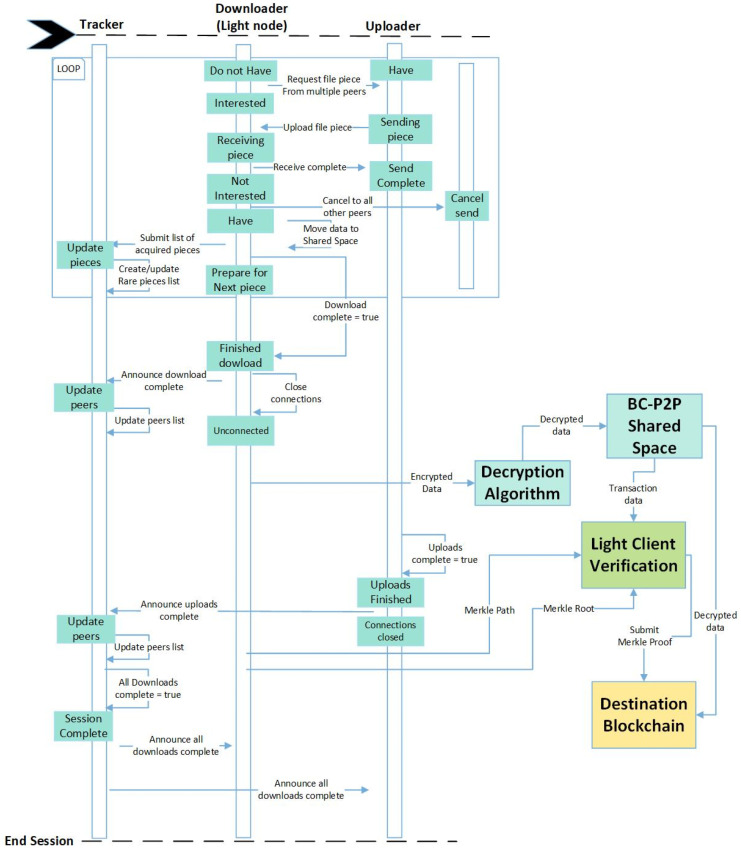
Interaction diagram part 2.

**Figure 5 sensors-24-07630-f005:**
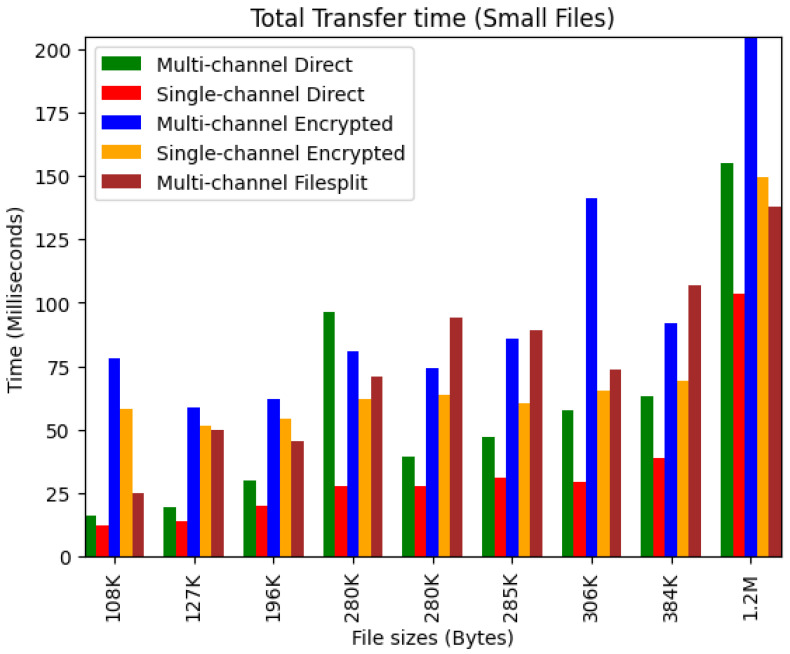
Small file transfer times.

**Figure 6 sensors-24-07630-f006:**
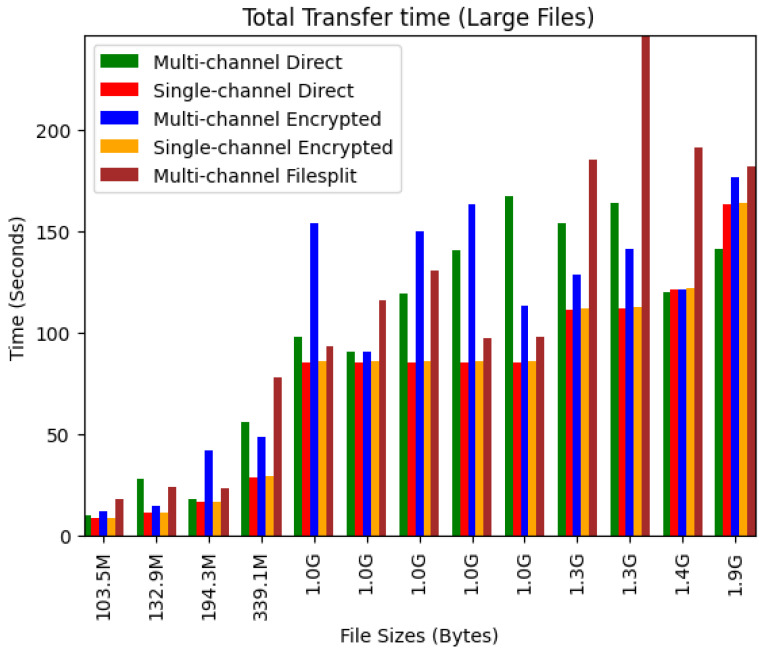
Large file transfer times.

**Figure 7 sensors-24-07630-f007:**
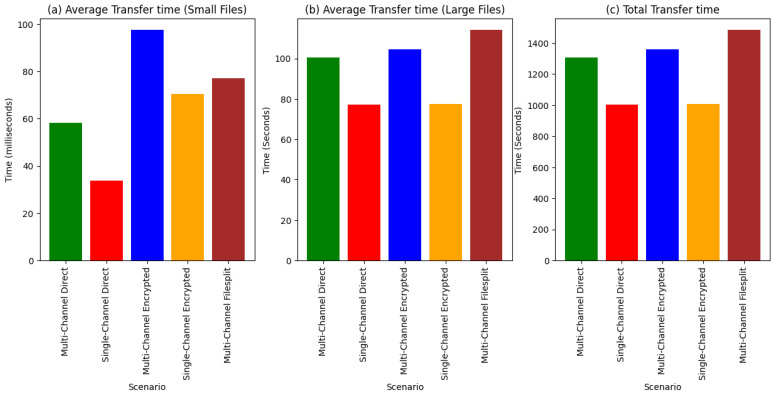
File transfer times for the given scenarios: (**a**) Average time it takes to transfer a small file per test scenario (**b**) Average time it takes to transfer a large file per test scenario (**c**) Total time it takes to transfer all files per test scenario.

## Data Availability

The raw data supporting the conclusions of this article will be made available by the authors on request.

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
