# Peer review of "A Framework for Full Decentralization in Blockchain Interoperability"

_sensors, 2024, doi:10.3390/s24237630_

Round 1
Reviewer 1 Report (Previous Reviewer 1)
Comments and Suggestions for Authors
Now contents are fine. Authors have made all the suggested chances. Hence manuscript is accepted in its present form.
Author Response
Please see the attachment.

Reviewer 2 Report (New Reviewer)
Comments and Suggestions for Authors
The authors present a scheme for allowing interoperability of blockchains. This is very imprortant. The paper is well-written and well-structured. The problem, the solution as well as a detailed architecture of the system is described. Some implementation issues are described. Some comparison to a single-client perr-to-peer is discussed.
Some minor issue to be fixed in the final version are as follows:
1- The captions of Figure 3 and Figure 4 shoudl be more specific, giving a hint on which part of the system interaction theu describe, instead of terming them as 1 and 2.
2- Also, the explaination of Figure 3 and 4 must be separated. First explain what the interation depicted in Figure 3 then, in a separate paragraph, explain the interation depicted in Figure 4.
3- I suggest to present Figure 7, 8 and 9 as subfigures side by side in the same figure. This is can be done clearly by reducing the width of the bars, which are now extremely large and wasting space in the paper with no improvement of clarity whatsoever.
4- In the conclusion, the authors must also discuss the limitation of the offer architecture and interation schemes. These must have some limitations surely.
Author Response
Please see the attachment.

This manuscript is a resubmission of an earlier submission. The following is a list of the peer review reports and author responses from that submission.
Round 1
Reviewer 1 Report
Comments and Suggestions for Authors
Please see attachment

Reviewer 2 Report
Comments and Suggestions for Authors
This paper proposes a heterogenous blockchain interoperability architecture.
This paper is more like a project report, and the scientific novelty is unclear.
The paper lacks a comparison study and numerical results showing the performance of the proposed scheme.
Reviewer 3 Report
Comments and Suggestions for Authors
The article concerns an interesting and up-to-date topic of the interoperability of various blockchain networks.
Unfortunately, the authors have not finished the article.
In my opinion, they reviewed the literature, selected algorithms and proven techniques from existing works, and compiled a concept of the architecture of a potential solution. Besides, the manner of modeling this architectural concept requires thorough refinement. However, this is where the article ends. There is no design, let alone implementation. So the results from running such a system are missing. There is also no verification. Theoretical verification/validation may involve equations, calculations, results, and proofs.
I encourage the authors to design at least a proof-of-concept and conduct performance tests on such an environment. Only with the design and implementation, real problems will emerge and assumptions can be verified.
I recommend rejecting the manuscript in its current form.